# Diagnostic Value of Fasting and Bedtime Saliva Pepsin Measurements in Laryngopharyngeal Reflux

**DOI:** 10.3390/biomedicines12020398

**Published:** 2024-02-08

**Authors:** Jerome R. Lechien, Francois Bobin

**Affiliations:** 1Division of Laryngology and Broncho-Esophagology, Department of Otolaryngology-Head Neck Surgery, EpiCURA Hospital, B7000 Baudour, Belgium; 2Department of Human Anatomy and Experimental Oncology, Faculty of Medicine, UMONS Research Institute for Health Sciences and Technology, University of Mons (UMons), Avenue du Champ de Mars, 6, B7000 Mons, Belgium; 3Department of Otorhinolaryngology and Head and Neck Surgery, Foch Hospital, School of Medicine, UFR Simone Veil, Université Versailles Saint-Quentin-en-Yvelines (Paris Saclay University), 92150 Paris, France; 4Department of Otorhinolaryngology and Head and Neck Surgery, CHU de Bruxelles, CHU Saint-Pierre, School of Medicine, F64000 Brussels, Belgium; 5Polyclinique Elsan de Poitiers, 86000 Poitiers, France; bobinfr@wanadoo.fr

**Keywords:** laryngopharyngeal, reflux, otolaryngology, pharyngeal, saliva, pepsin

## Abstract

Background: The pepsin test is an emerging non-invasive diagnostic approach for laryngopharyngeal reflux (LPR). The aim of this study was to investigate the diagnostic value of multiple salivary pepsin tests for detecting LPR. Methods: Patients with suspected LPR and asymptomatic individuals were consecutively recruited from January 2020 to November 2022. Patients benefited from hypopharyngeal–esophageal impedance-pH monitoring (HEMII-pH) and fasting and bedtime saliva collections to measure oral pepsin. The sensitivity, specificity, and positive (PPV) and negative (NPV) predictive values were calculated considering fasting, bedtime, and the highest values of the pepsin tests at ≥16, ≥36, ≥45, and ≥100 ng/mL cutoffs. Results: The pepsin test was adequately performed in 147 LPR patients and 32 controls. The pepsin tests were 81.6%, 74.8%, and 61.5% sensitive at cutoffs of ≥16, ≥45, and ≥100 ng/mL, respectively. The PPVs were 93.0%, 94.0%, and 94.8%, respectively. The highest specificity (81.8%) was found for the fasting pepsin test at a cutoff of 100 ng/mL. The highest sensitivity (81.6%) was found by considering the highest measured pepsin test at the ≥16 ng/mL threshold. The measurement of fasting saliva pepsin was associated with the highest sensitivity and specificity value. At ≥16 ng/mL, 27 patients had negative findings, indicating that 18.4% (27/147) of the true positive cases were missed by considering the highest pepsin test. The receiver operating characteristic curve reported that a cutoff of 21.5 was 76.9% sensitive and 62.5% specific, while the PPV and NPV were 91.1% and 38.2%, respectively. Conclusions: The consideration of the highest concentration of the fasting and bedtime saliva pepsin collections at a cutoff of 21.5 was associated with the best detection rate and sensitivity of the pepsin tests.

## 1. Introduction

Laryngopharyngeal reflux (LPR) is an inflammatory condition of the upper aerodigestive tract tissues related to direct and indirect effects of gastroduodenal content reflux, which induces morphological changes in the upper aerodigestive tract [1]. The deposit of pepsin into the upper aerodigestive tract mucosa leads to the development of an inflammatory reaction and related symptoms and signs [2,3]. To date, the diagnosis may be confirmed through the identification of acid, weakly acid, and nonacid esophago-pharyngeal reflux events at the 24 h hypopharyngeal–esophageal multichannel intraluminal impedance-pH monitoring (HEMII-pH) [3]. HEMII-pH is, however, little used by the majority of American and European otolaryngologists [4,5] because this approach is inconvenient and some patients do not tolerate the probe [6]. Over the past two decades, oral salivary pepsin detection (pepsin test) has been suggested as an alternative non-invasive diagnostic modality [7]. The saliva pepsin concentration may be measured in a consultation office but there is no consensus regarding the most appropriate time for saliva collection (fasting post-meal versus bedtime), and the adequate number of samples to make the LPR diagnosis. Moreover, several cutoffs of pepsin concentration have been proposed by authors without a consensus being reached [7,8,9]. The main limitations of previous studies were the lack of inclusion of asymptomatic individuals, which may be associated with inaccuracy of specificity (SP) and predictive value assessments [10]. Moreover, most authors did not assess the sensitivity (SE), SP, and predictive values of several pepsin test thresholds [10].

The aim of this study was to investigate the diagnostic value of fasting and bedtime salivary pepsin tests for detecting laryngopharyngeal reflux.

## 2. Material and Methods

### 2.1. Subjects and Setting

Patients with LPR symptoms and findings and asymptomatic individuals were consecutively recruited from two hospitals (Elsan Polyclinic of Poitiers, France; CHU Saint-Pierre, Brussels, Belgium) from January 2020 to November 2022. The diagnosis was confirmed through positive HEMII-pH. A gastrointestinal (GI) endoscopy was proposed to elderly patients and those with gastroesophageal reflux disease (GERD) symptoms. Asymptomatic individuals were recruited if they reported a Reflux Symptom Score-12 (RSS-12) < 11 [11] and Reflux Sign Assessment (RSA) < 14 [12], and no exclusion criteria similar to LPR patients. Individuals with negative HEMII-pH were also considered as ‘controls’. The exclusion criteria included smoker, alcohol dependence, neurological or psychiatric illness, upper respiratory tract infection within the last month, current use of anti-reflux treatment or inhaled corticosteroids, previous history of neck surgery or trauma, benign vocal fold lesions, malignancy, history of ear, nose, and throat radiotherapy, and active seasonal allergies or asthma [3]. The local ethics committee approved the study protocol (n°BE076201837630). The patients consented to participate.

### 2.2. Hypopharyngeal–Esophageal Multichannel Intraluminal Impedance-pH Monitoring

The catheter model used was introduced transnasally and the length was chosen based on the esophageal length of patient. The catheter was placed in the morning before breakfast (8:00 a.m.) at the hospital by an experienced practitioner. The HEMII-pH was composed of 8 impedance segments and 2 pH electrodes (Versaflex Z^®^, Digitrapper pH-Z testing System, Medtronic, Lille, France, Europe). The six esophageal impedance segments were placed along the esophagus zones (Z1 to Z6) at 19, 17, 11, 9, 7, and 5 cm above the lower esophageal sphincter (LES). The pharyngeal impedance segments were placed 1 and 2 cm above the cricopharyngeal sphincter in the hypopharynx. The pH electrodes were placed 2 cm above the LES and 1 to 2 cm below the cricopharyngeal sphincter, respectively. The LPR diagnosis was based on the occurrence of a >1 acid or nonacid hypopharyngeal reflux event, which was defined as an episode reaching the two impedance sensors in the hypopharynx [13]. An acid reflux episode consisted of an episode with pH ≤ 4.0. A nonacid reflux episode consisted of an episode with pH > 4.0. Patients were taken off of proton pump inhibitors during the HEMII-pH testing. Patients were instructed to maintain their usual daily diet, activities, and non-reflux medications during the 24 h recording.

### 2.3. Saliva Pepsin Measurement

Patients and healthy individuals collected saliva samples (1 to 5 mL) in the morning (fasting, after waking) and 2 h after dinner (bedtime) during the 24 h HEMII-pH period. The saliva was collected into a 30 mL universal sample collection tube containing a pre-established concentration of citric acid to preserve the action of any pepsin present (Peptest^®^ kit, RD Biomed Ltd., Hull, UK). The pepsin sample collections were stored in the refrigerator. The measurement of the pepsin concentration in the saliva samples was performed through a Peptest^®^ device (RD Biomed Ltd., Hull, UK). The steps of pepsin measurement were performed in a standardized procedure, which has been previously described [14]. The pepsin measurement was performed by an experienced practitioner in a blind manner. The saliva pepsin concentration was measured using the Cube Reader^®^ and ranged from 1 to 500 ng/mL.

### 2.4. Demographics, Symptoms, and Signs

Demographic data, i.e., age, gender, height, weight, body mass index, were collected from patient records. The symptoms and findings were assessed with RSS-12 [11] and RSA [12], respectively. RSS-12 is a self-administered validated 12-item reported-outcome questionnaire documenting the frequency and severity of ear, nose, throat, digestive, and respiratory complaints. RSA is a validated 61-point finding score assessing laryngeal and extra-laryngeal findings.

## 3. Statistical Methods

Statistical analyses were performed using the Statistical Package for the Social Sciences for Windows (SPSS version 27.0; IBM Corp, Armonk, NY, USA). A power analysis was performed, in which the ideal sample size for our study was calculated focusing on the diagnostic accuracy of pepsin tests in the previous studies. Precisely, the anticipated SE of the tests was set at 85.0, indicating an expectation of a high true positive rate, while the anticipated SP was set at 40.0, acknowledging a relatively high rate of false positives. We assumed an imbalance in the distribution of healthy to diseased individuals, with a ratio (R) set to 1/5, reflecting the prevalence of LPR in the population (10%). The statistical power, a measure of the study’s ability to detect a true effect, was set at a standard value of 0.80. The significance level, a threshold for determining statistical significance, was set at 0.05. Based on these assumptions, a simplified R function was used to calculate the required sample size.

The SE, SP, and positive (PPV) and negative predictive values (NPV) of several pepsin saliva cutoffs (≥16, ≥36, ≥45, and ≥100 ng/mL) were calculated for fasting, bedtime, and the highest concentrations of both pepsin saliva samples (fasting and bedtime). A multivariate analysis was used to study the relationships between the HEMII-pH findings, the pepsin saliva concentration, and the clinical features. The association was considered as low, moderate, or strong for r_s_ < 0.30, 0.30–0.60, or r_s_ > 0.60, respectively. The consistency between HEMII-pH and the pepsin test was assessed with the kappa-Cohen analysis. The pepsin test cutoff for determining the presence and absence of LPR was examined using a receiver operating characteristic (ROC) analysis.

## 4. Results

Of the 174 patients who underwent 24 h HEMII-pH, the LPR diagnosis was confirmed in 163 cases. Sixteen patients with positive HEMII-pH were excluded because of unreadable morning and bedtime pepsin tests. The control group was composed of 11 individuals without LPR at the 24 h HEMII-pH, and 21 asymptomatic individuals with RSS-12 < 11 and RSA < 14. The asymptomatic subjects were recruited from the University of Mons. The mean age of the LPR patients was 53.8 ± 14.5 years. There were 90 females (61.2%) and 57 males (38.8%). A GI endoscopy was performed in 107 (72.8%) patients and was normal in 34 (31.8%) patients (Table 1). The mean RSS-12 and RSA were 64.9 ± 49.2 and 21.9 ± 8.5, respectively. The control group included 21 females and 11 males, with ages ranging from 18 to 69 years old. No one had a history of GERD or LPR. Pepsin was undetectable in 17 of the asymptomatic/control individuals (53.1%), while 15 subjects reported at least one pepsin concentration > 16 ng/mL.

The accuracy of the pepsin test according to various cutoffs is reported in Table 2. The morning pepsin test detected LPR in 43.3% to 66.0% of cases. The bedtime pepsin test was found to detect LPR in 43.8% to 63.7% of cases regarding pepsin saliva cutoffs. The highest accuracy values were obtained by considering the highest pepsin saliva concentration of both pepsin tests for cutoffs ≥16 and ≥36 (Table 2).

The SE, SP, PPVs, and NPVs of the pepsin tests at cutoffs ≥16, ≥36, ≥45, and ≥100 ng/mL are reported in Table 3. The salivary pepsin test was 61.5%, 63.9%, and 81.6% sensitive at cutoffs ≥16 ng/mL when considering fasting, bedtime, and the highest concentration of both pepsin tests, respectively. The PPVs were 91.3%, 92.0%, 93.0%, respectively. The highest SP (81.8%) was found for the fasting pepsin test at a cutoff of ≥100 ng/mL (Table 3).

The ROC curve is shown in Figure 1. A threshold of 21.5 was 76.9% sensitive and 62.5% specific. The PPV was 91.1%, whereas the NPV was 38.2%.

The consistency analysis between HEMII-pH and the pepsin tests was not significant. The only significant consistency results concerned the consistencies between a HEMII-pH positive diagnosis with RSA > 14 (kappa = 0.422, *p* = 0.001) or RSS-12 > 11 + RSA > 14 (kappa = 0.382; *p* = 0.002). There were no significant associations between levels of salivary pepsin and age, sex, or body mass index. There was a poor but significant association between the number of acid pharyngeal reflux events and the mean concentrations of pepsin tests (r_s_ = 0.163; *p* = 0.041). The significant association between the number of acid pharyngeal reflux events and the RSA was moderate (r_s_ = 0.434; *p* = 0.001). The bedtime saliva pepsin concentration was significantly associated with the RSA score (r_s_ = 0.368; *p* = 0.009).

## 5. Discussion

The clinical diagnosis of LPR is challenging regarding the non-specificity of symptoms and signs and the low therapeutic success rate of empirical treatment [15,16,17,18,19]. HEMII-pH is considered as the most reliable tool, but this approach remains expensive and poorly tolerated by patients [8]. In this context, many authors have tried to develop non-invasive objective clinical tools for detecting LPR. The pepsin saliva test is one of the most studied clinical tools over the past few years, but there is no consensus about the number of samples, the cutoffs, and the time of saliva collection [2,8,9,10,20,21]. These parameters have been investigated in several studies in which the authors have suggested most frequently ≥16, ≥45, or ≥100 ng/mL pepsin test thresholds (Table 4) [7,8,9,20,21,22,23,24,25,26]. The findings of the present study supported that a cutoff for the pepsin test of ≥16 ng/mL, which was measured on the highest pepsin tests (fasting and bedtime), was associated with the highest diagnosis rate and SE. Thus, as supported by Wang et al. [20], the realization of several pepsin saliva collections is necessary to improve the accuracy, SE, and PPV of the test. Wang et al. [20] observed in 97 LPR patients that 55.7% of the true positive cases were missed by considering a single pepsin test.

The SE and PPV findings of the present study corroborated the trends observed in the recent study of Zhang et al. [8], who observed SE values of 57.7% at a cutoff of ≥16 ng/mL and PPV of 87.0. However, the authors reported higher values of SP, which were 25.0% at a cutoff ≥16 ng/mL [8]. Interestingly, Zhang et al. suggested in a cohort of 35 patients that the collection of saliva during symptomatic periods within the testing day conferred superior SP at ≥16 ng/mL (66.7%). This observation could explain the potential differences in SP with our study because our patients were not particularly symptomatic at the time of saliva collection (fasting and bedtime). However, note that the collection of saliva during the symptom periods remains controversial. Indeed, Na et al. reported better values of SE and SP when the pepsin tests were performed post-meals compared to pepsin tests at the symptom occurrence [27]. Compared to the data of Zelenik et al. [9], our study reported higher SE and PPV values, while Weitzendorfer et al. [23] found higher SE values compared to our data (Table 4). Regarding the controversies in the literature about the cutoff, we used the ROC curve to determine the threshold associated with the highest SE and SP values. A cutoff of 21.5 was found to be associated with an SE and SP of 76.9% and 62.5%. To the best of our knowledge, this is the first study using the ROC curve to determine a specific threshold for the pepsin test, while the other studies only selected arbitrary thresholds to assess SE, SP, and predictive values.

The fasting saliva collection (morning, ≥16 ng/mL) reports a slightly higher detection rate, SE, SP, PPV, and NPV. Na et al. observed similar findings in a cohort of 50 LPR patients [27]. The superiority of the morning-fasting pepsin saliva measurement over the bedtime/evening assessment was supported in the study of Wang et al., who observed the highest accuracy of the pepsin test at 7:00 and 8:00 a.m. [20]. Moreover, the authors reported that the salivary pepsin (cutoff ≥ 45 ng/mL) detected in the morning had an SE of 38.4% and an SP of 84.6% for the diagnosis of laryngopharyngeal reflux. From a diagnostic standpoint, we did not find significant consistency between pepsin tests and HEMII-pH. Focusing on the HEMII-pH data, the number of acid pharyngeal reflux events was associated with the mean concentration of saliva pepsin. These observations contributed to the controversy regarding the correlation between HEMII-pH findings and pepsin saliva measurements. Indeed, some authors have reported low-to-moderate significant associations [8,20], while others have not found significant association [28]. To date, there is no agreed-upon cutoff for salivary pepsin as a diagnostic marker in LPR because of variability in the levels of saliva throughout the testing day [10]. Thus, the numerous inconsistencies between studies [8,9,10,20] may highlight the variability in the saliva pepsin concentration throughout the testing day across individuals [20]. Currently, the pattern of the variation in the saliva pepsin concentration and the related influencing factors are poorly understood. In a recent prospective study, it was suggested that foods and beverages significantly influenced the saliva pepsin concentration of patients with LPR [29]. The differences across studies from different world regions should be interpreted considering the dietary habits of populations. The use of the diet refluxogenic score, assessing the refluxogenic potential of foods and beverages [29], may be considered in studies evaluating the pepsin saliva concentration over time.

The consideration of diet during the testing period may explain the findings of the control group. Indeed, an important issue highlighted in the present study was the number of individuals with at least one positive pepsin test without positive LPR diagnosis at the HEMII-pH. Of the 11 patients without LPR at the HEMII-pH, 9 had at least one positive pepsin test at a threshold of 16 ng/mL. The presence of pepsin in the mucosa of asymptomatic or healthy individuals may strengthen the argument for the role of diet in the occurrence of transient esophageal sphincter relaxations, and the related esophago-pharyngeal reflux events [30]. Because pepsin may be internalized in laryngopharyngeal cells, and externalized in a second time [31], the pepsin test could be theoretically positive outside of the 24 h HEMII-pH period. This issue, as well as the lack of symptoms in individuals with positive pepsin tests, needs future studies.

Another factor that may explain the inconsistencies in the associations between HEMII-pH, pepsin saliva measurements, and clinical findings is the potential contribution of other gastroduodenal enzymes in the development of the LPR inflammatory reaction and the development of related symptoms and signs. Indeed, pepsin is probably not the only gastroduodenal enzyme able to irritate the laryngopharyngeal mucosa. Bile salts were found in patients with LPR [28,32] and appear to be involved in laryngeal and pharyngeal inflammation and related cancer [33,34,35,36,37].

To the best of our knowledge, the present study is the largest cohort study investigating the SE, SP, PPV, NPV, and accuracy of the pepsin saliva test, which is the main strength of our study. As for the other recent studies [8,9,20], the main limitation of the present study is the homogeneity of the study population, which mainly included patients with a positive diagnostic at the HEMII-pH. The very low number of individuals with a negative HEMII-pH may lead to inaccuracy in the SP and NPV assessments. For this reason, we tried to limit the negative effect of this bias through the inclusion of 21 asymptomatic individuals without significant symptoms and findings, which improved the SP and NPV calculations. However, these additional controls did not undergo HEMII-pH because of the cost and inconvenience, which is a limitation regarding the lack of association between symptoms, findings, and the data at the HEMII-pH [15,38,39,40,41,42,43].

## 6. Conclusions

The consideration of the highest concentration from fasting and bedtime saliva pepsin collections is associated with the best detection rate and sensitivity. A threshold of 21.5 was 76.9% sensitive and 62.5% specific. Depending on the practitioner’s wish, superior specificities conferring greater diagnostic value may be achieved using higher thresholds for the pepsin saliva test. Future studies are needed to confirm the most adequate number and time of saliva sample collection for the pepsin measurement.

## Figures and Tables

**Figure 1 biomedicines-12-00398-f001:**
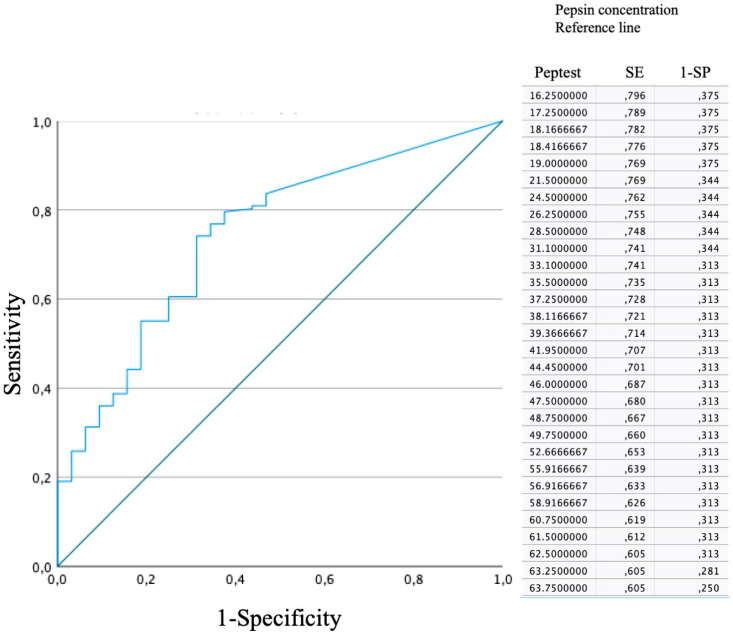
Receiver operating characteristic curve. Considering the highest pepsin concentration, a threshold of 21.5 was 76.9% sensitive and 62.5% specific.

**Table 1 biomedicines-12-00398-t001:** Characteristics of patients.

Characteristics	
Age (mean, SD)	53.8 ± 14.5
BMI (mean, SD)	26.7 ± 6.2
Male (N, %)	57 (38.8)
Female (N, %)	90 (61.2)
Gastrointestinal endoscopy (N = 107)	
Normal	34 (31.8)
Esophagitis	12 (11.2)
Hiatal hernia	43 (40.2)
LES insufficiency	45 (42.1)
Gastritis	11 (10.3)
HEMII-pH (mean, SD)	
Pharyngeal acid events	8.4 ± 13.3
Pharyngeal nonacid events	17.3 ± 32.0
Pharyngeal events (total number)	25.7 ± 33.3
Clinical data	
Reflux Symptom Score-12 (mean, SD)	64.9 ± 49.2
Reflux Sign Assessment (mean, SD)	
Oral score	5.3 ± 2.4
Pharyngeal score	9.4 ± 4.2
Laryngeal score	8.6 ± 5.0
Reflux Sign Assessment	21.9 ± 8.5

Abbreviations: BMI = body mass index; HEMII-pH = hypopharyngeal–esophageal multichannel intraluminal impedance-pH monitoring; LES = lower esophageal sphincter; N = number; SD = standard deviation.

**Table 2 biomedicines-12-00398-t002:** Accuracy of saliva pepsin test according to thresholds in patients with reflux at the pH-impedance testing.

Pepsin Tests	Accuracy
Morning (N = 141)	
≥16 ng/mL	66.0
≥36 ng/mL	56.7
≥45 ng/mL	55.3
≥100 ng/mL	43.3
Bedtime (N = 146)	
≥16 ng/mL	63.7
≥36 ng/mL	58.2
≥45 ng/mL	55.5
≥100 ng/mL	43.8
Highest sample concentration (N = 147)	
≥16 ng/mL	81.6
≥36 ng/mL	77.5
≥45 ng/mL	74.8
≥100 ng/mL	61.5

**Table 3 biomedicines-12-00398-t003:** Characteristics of patients according to the reflux profiles.

	Morning Pepsin Test				Bedtime Pepsin Test				Highest Pepsin Test			
	SE	SP	PPV	NPV	SE	SP	PPV	NPV	SE	SP	PPV	NPV
≥16 ng/mL	65.1	62.5	88.8	28.2	63.9	64.5	89.3	27.8	81.6	53.1	88.9	38.6
≥36 ng/mL	56.8	78.1	92.2	28.4	57.6	74.1	91.2	27.4	77.5	65.6	91.2	38.9
≥45 ng/mL	52.7	81.8	92.8	28.1	56.2	75.0	91.0	27.6	74.8	70.6	91.7	39.3
≥100 ng/mL	42.2	86.2	93.9	22.7	44.8	84.4	92.9	25.2	61.5	78.1	92.9	30.5

Abbreviations: SE = sensitivity; SP = specificity; PPV = positive predictive value; NPV = negative predictive value.

**Table 4 biomedicines-12-00398-t004:** Literature studies.

Author	Year	LPR/CT	Diagnosis of LPR	Sample Time	Pepsin Analysis	Cutoffs	SE	SP	PPV	NPV	Accuracy
Wang [18]	2022	112/26	HEMII-pH (Sandhill);	Each hour from	Pepsin lateral	>45 ng/mL	38.4	84.6	91.5	24.2	55.7
			≥1 pharyngeal event	7:00 a.m. (fasting)	flow device;	Fasting test					
				to 6:00 p.m.	highest of	>45 ng/mL	86.6	80.8	95.1	58.3	73.9
					concentrations	Highest test					
					in morning						
Zelenik [9]	2021	45/1	HEMII-pH (Medtronic);	Fasting	PepTest	≥16 ng/mL	48.0	27.0	63.0	40.0	48.0
			>1 pharyngeal event								
Zhang [8]	2020	26/4	HEMII-pH (Sandhill);	Fasting;	PepTest	≥16 ng/mL	76.9	25.0	87.0	14.3	87.0
			≥2 pharyngeal events,	1 h post-lunch,		≥75 ng/mL	57.7	75.0	93.8	21.4	N.P.
			≥6 proximal events	1 h post-dinner,							
				when symptoms							
Klimara [19]	2019	19/7	HEMII-pH (Sandhill);	Fasting;	ELISA	>1 ng/mL	29.4	50.0	62.5	20.0	42.0
			>1 pharyngeal event,	1 h post-lunch,	Western blot;						
			>40 proximal events	1 h post-dinner,	highest						
				1 h post-breakfast	concentration						
Weitzendorfer	2019	41/29	Oropharyngeal pH test;	3 samples	PepTest;	>16 ng/mL	85.4	27.6	62.5	57.1	85.4
[20]			Ryan score >9.4		highest of	>50 ng/mL	78.1	41.4	65.3	57.1	
					3 samples	>100 ng/mL	68.3	58.6	70.0	56.7	
						>150 ng/mL	53.7	69.0	71.0	51.3	
						>216 ng/mL	41.5	86.2	81.0	51.0	
Hayat [7]	2015	111/100	MII-pH;	Fasting;	PepTest	>16 ng/mL	77.6	63.2	58.4	80.4	N.P.
			esophageal acid	1 h post-lunch,		>50 ng/mL	67.2	76.3	67.2	76.8	
			exposure time pH < 4,	1 h post-dinner		>100 ng/mL	51.7	74.5	54.5	72.3	
			>4.2%			>150 ng/mL	41.4	90.8	75.0	69.9	
						>210 ng/mL	44.2	96.3	95.7	36.5	
Ocak [21]	2015	18/2	Dual-probe pH testing;	N.P.	PepTest	≥16 ng/mL	33.0	100	100	14.2	N.P.
			distal esophageal pH								
			time pH < 4.0, >5%								
Saritas [22]	2012	22/25	Wireless pH testing;	N.P.	PepTest	≥50 ng/mL	50.0	92.0	85.0	68.0	N.P.
			esophageal acid								
			exposure time pH < 4,								
			>4.2%								
Potluri [22]	2003	3/13	Dual-probe pH testing;	When symptoms	Pepsin	>1 ng/mL	100	92.3	N.P.	N.P.	N.P.
			≥1 proximal acid		assay						
			esophageal event								

Abbreviations: (HE)MII-pH = hypopharyngeal–esophageal multichannel intraluminal impedance-pH monitoring; LPR/CT = laryngopharyngeal reflux/controls; NP = not provided; NPV = negative predictive value; PPV = positive predictive value; SE = sensitivity; SP = specificity.

## Data Availability

Data are available on request.

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
