# Peer review of "Diagnostic Value of Fasting and Bedtime Saliva Pepsin Measurements in Laryngopharyngeal Reflux"

_biomedicines, 2024, doi:10.3390/biomedicines12020398_

Round 1

Reviewer 1 Report

Comments and Suggestions for Authors

The authors focused their study on the diagnostic parameters of the salivary pepsin test. Indeed, diagnosing laryngopharyngeal reflux is a difficult task. Currently, 24-h dual-probe esophageal pH monitoring or impedance is considered the best diagnostic methods, but they have many disadvantages. Detection of pepsin in saliva would be a very good diagnostic test. However, there are a number of issues with its use. Although there are a number of similar studies, the authors obtained new information regarding the timing of saliva collection. The research is of a high scientific standard, carefully thought out and excellently presented. While reading the text, I only had a few small questions.

1. In the abstract (lines 34 and 36), please add ng/mL after 21.5.

2. It seems to me that patients with dental problems (stomatitis, gingivitis, etc.) should be added to the exclusion criteria. I think the authors actually excluded such patients. The inflammatory process in the oral cavity will change the composition of saliva, which may affect the pepsin content. I think this point needs to be pointed out.

3. The main result of the study are Table 2 and 3, where accuracy, SE, SP, PPV and NPV are given for the four cutoff points. However, it seems to me that more information can be extracted from the data obtained by the authors. I would advise the authors to consider expanding the results and including positive likelihood ratio (PLR), negative likelihood ratio (NLR), and maybe Youden’s index and predictive summary index (PSI).

4. It seems to me that it is better to change the name of table 3. Instead of “Characteristics of Patients according to the reflux profiles,” it is better to call it “Parameters of the diagnostic test for pepsin in saliva at different cutoffs” (or something like this).

Author Response

Reviewer 1:

The authors focused their study on the diagnostic parameters of the salivary pepsin test. Indeed, diagnosing laryngopharyngeal reflux is a difficult task. Currently, 24-h dual-probe esophageal pH monitoring or impedance is considered the best diagnostic methods, but they have many disadvantages. Detection of pepsin in saliva would be a very good diagnostic test. However, there are a number of issues with its use. Although there are a number of similar studies, the authors obtained new information regarding the timing of saliva collection. The research is of a high scientific standard, carefully thought out and excellently presented. While reading the text, I only had a few small questions.

Thank you.

1.In the abstract (lines 34 and 36), please add ng/mL after 21.5.

Thank you. We corrected: line 34: “Receiver operating characteristic curve reported that a cutoff of 21.5 ng/mL was 76.9% sensitive and 62.5% specific, while the PPV and NPV were 91.1% and 38.2%, respectively.”

  1. It seems to me that patients with dental problems (stomatitis, gingivitis, etc.) should be added to the exclusion criteria. I think the authors actually excluded such patients. The inflammatory process in the oral cavity will change the composition of saliva, which may affect the pepsin content. I think this point needs to be pointed out.

Yes, we excluded them. We corrected: methods: p.4, last line: “The exclusion criteria included: smoker, alcohol dependence, neurological or psychiatric illness, upper respiratory tract infection within the last month, current use of anti-reflux treatment or inhaled corticosteroids, previous history of neck surgery or trauma, benign vocal fold lesions, malignancy, history of ear, nose, and throat radiotherapy, stomatitis, gingivitis, and active seasonal allergies or asthma [3].”

  1. The main result of the study are Table 2 and 3, where accuracy, SE, SP, PPV and NPV are given for the four cutoff points. However, it seems to me that more information can be extracted from the data obtained by the authors. I would advise the authors to consider expanding the results and including positive likelihood ratio (PLR), negative likelihood ratio (NLR), and maybe Youden’s index and predictive summary index (PSI).

We added LH+, LH-, and DSI in Table 3, page 16, and in the results.

  1. It seems to me that it is better to change the name of table 3. Instead of “Characteristics of Patients according to the reflux profiles,” it is better to call it “Parameters of the diagnostic test for pepsin in saliva at different cutoffs” (or something like this).

We have changed using the title suggested by the reviewer.

Reviewer 2 Report

Comments and Suggestions for Authors

Diagnostic Value of Fasting and Bedtime Saliva Pepsin Measurements in Laryngopharyngeal Reflux

Abstract

35 Conclusion: The consideration of the highest concentration of fasting and bedtime saliva pepsin collections at cutoff 21.5 was associated with the best detection

36 rate and sensitivity of pepsin tests.

Comment: In result section no information about bedtime saliva concentration

71 exclusion criteria included: smoker, alcohol dependence, neurological or psychiatric ill-

72 ness, upper respiratory tract infection within the last month

Comment: What about acute gastrointestinal diseases

66 The diagnosis was con- 66 firmed through positive HEMII-pH.

87 diagnosis was based on the occurrence of >1 acid or nonacid hypopharyngeal reflux event, 88 which was defined as an episode reaching the two impedance sensors in the hypopharynx 89 [13].

Comment: How accurate the LPR diagnosis when >1, according to [13]. Was it enough to define LPR relaying only on HEMII-pH  

96 (fasting, after waking) and 2 hours after the dinner (bedtime) during the 24-hour

Comment: lying or upright position

100 The pepsin sample collections were stored in the refrigerator

Comment: At what T, for how long ?

No clear definition how was selected control group and asymptomatic ? Why asymptomatic were included as control ? Control and asymptomatic are two groups but in the results they are one control group ?

Table 1. Characteristics of Patients.

Gastrointestinal endoscopy (N=107)

Normal 34 (31.8 what it is ? )

The 153 morning pepsin test detected LPR in 43.3% to 66.0% of cases. The bedtime pepsin test was 154 found to detect LPR in 43.8% to 63.7% regarding pepsin saliva cutoffs. The highest

Comment: what is the correlation in the same patient between morning and bedtime saliva pepsin.

What is the correlation of symptoms with saliva pepsin in the same patient.

Comments on the Quality of English Language

No comments

Author Response

Reviewer 2

Abstract

Conclusion: The consideration of the highest concentration of fasting and bedtime saliva pepsin collections at cutoff 21.5 was associated with the best detection rate and sensitivity of pepsin tests.

We added: Results, paragraph 2: line 2: “The bedtime pepsin test was found to detect LPR in 43.8% to 63.7% regarding pepsin saliva cutoffs.”

71 exclusion criteria included: smoker, alcohol dependence, neurological or psychiatric ill-

72 ness, upper respiratory tract infection within the last month …

Comment: What about acute gastrointestinal diseases

We excluded them. We specified: Methods, exclusion criteria: line 5: “The exclusion criteria included: smoker, alcohol dependence, neurological or psychiatric illness, upper respiratory tract or gastrointestinal infection within the last month, current use of anti-reflux treatment or inhaled corticosteroids, previous history of neck surgery or trauma, benign vocal fold lesions, malignancy, history of ear, nose, and throat radiotherapy, stomatitis, gingivitis, and active seasonal allergies or asthma [3].”

66 The diagnosis was con- 66 firmed through positive HEMII-pH.

87 diagnosis was based on the occurrence of >1 acid or nonacid hypopharyngeal reflux event, 88 which was defined as an episode reaching the two impedance sensors in the hypopharynx 89 [13].

Comment: How accurate the LPR diagnosis when >1, according to [13]. Was it enough to define LPR relaying only on HEMII-pH  

Yes, we based them of the Dubai Consensus. In the Dubai consensus, authors supported the diagnosis when there are >1 HRE. A systematic review of normative value of HEMII-pH supported also this criteria. doi: 10.1177/01945998211029831.

To improve clarity, we replaced the ref. 13 (normative data review) with Dubai Consensus.

96 (fasting, after waking) and 2 hours after the dinner (bedtime) during the 24-hour

Comment: lying or upright position

We specified: upright.

Methods, saliva pepsin measurement, p.5, line 1: “Patients and healthy individuals collected saliva samples (1 to 5 mL) in the morning (fasting, after waking, upright) and 2 hours after the dinner (bedtime) during the 24-hour HEMII-pH period.”

100 The pepsin sample collections were stored in the refrigerator

Comment: At what T, for how long ?

We specified: “The pepsin sample collections were stored in the refrigerator (0°C) less than 1 week.”

No clear definition how was selected control group and asymptomatic ? Why asymptomatic were included as control ? Control and asymptomatic are two groups but in the results they are one control group ?

This is the same group of asymptomatic patients (some having HEMII-pH, other no), we just used 2 different terms. Previous studies have showed that “control patients or asymptomatic individuals” need to report RSS-12<11 and RSA<14.

We developed in Methods, p.4, line 5: “Asymptomatic individuals were recruited from the University of Mons if they reported reflux symptom score-12 (RSS-12)<11 [11] and reflux sign assessment (RSA)<14 [12], and no exclusion criteria similar to LPR patients. Individuals with negative HEMII-pH were also considered as asymptomatic individuals if they reported RSS-12<11 and RSA<14.” 

Moreover, we changed in the manuscript the term “control” with asymptomatic individuals.

Table 1. Characteristics of Patients.

Gastrointestinal endoscopy (N=107)

Normal 34 (31.8 what it is ? )

We specified as requested: Table 1 footnotes: “Gastrointestinal results are found as Number (%).”

The 153 morning pepsin test detected LPR in 43.3% to 66.0% of cases. The bedtime pepsin test was 154 found to detect LPR in 43.8% to 63.7% regarding pepsin saliva cutoffs. The highest…

Comment: what is the correlation in the same patient between morning and bedtime saliva pepsin.

What is the correlation of symptoms with saliva pepsin in the same patient.

The multivariate analysis did not report significant correlations.

We specified the correlations, results, p.7: last paragraph: “There was a poor but significant association between the number of acid pharyngeal reflux event and the mean concentration of pepsin tests (rs=0.163; p=0.041). The significant association between the number of acid pharyngeal reflux event and the RSA was moderate (rs=0.434; p=0.001). The bedtime saliva pepsin concentration was significantly associated with the RSA score (rs=0.368; p=0.009).”

Reviewer 3 Report

Comments and Suggestions for Authors

L43: Is the deposit time is long, please mention the approximate deposit and the associated symptoms and sign.
L43:
Is the global incidence of LPR roughly known, i.e. how many percent of people have this condition?

Please state the factors cause the LPR briefly in the introduction.

L64: Age of patients should be stated here; are there any age-related symptoms of LPR?

L101: How long were samples in the refrigerator?

Table 4: Authors column should be moved to right.

Author Response

L43: Is the deposit time is long, please mention the approximate deposit and the associated symptoms and sign. 

The deposit time is not long since a HEMII-pH episode reach the pharynx in few seconds.
There was the following correlation: results, last paragraph: “We specified the correlations, results, p.7: last paragraph: “There was a poor but significant association between the number of acid pharyngeal reflux event and the mean concentration of pepsin tests (rs=0.163; p=0.041).”

L43: Is the global incidence of LPR roughly known, i.e. how many percent of people have this condition? Please state the factors cause the LPR briefly in the introduction. 

We added in the discussion the following sentence: Introduction, p.4, line 2: “The prevalence of LPR is still unknown but it seems that 10% to 30% of outpatients consulting in otolaryngology department reported LPR symptoms [1].”

L64: Age of patients should be stated here; are there any age-related symptoms of LPR?

We added: Methods, p.4, line 3: “The mean age of patients was 53.8 ± 14.5 years old.”

There was no significant correlation between age and symptoms but note that the cohort is small, which reduces the statistical power.

L101: How long were samples in the refrigerator? 

We specified: “The pepsin sample collections were stored in the refrigerator (0°C) less than 1 week.”

Table 4: Authors column should be moved to right.

Done.
